# Augmentative Message Passing for Traveling Salesman Problem and Graph Partitioning

**Siamak Ravanbakhsh**
Department of Computing Science
University of Alberta
Edmonton, AB T6G 2E8
mravanba@ualberta.ca

**Reihaneh Rabbany**
Department of Computing Science
University of Alberta
Edmonton, AB T6G 2E8
rabbanyk@ualberta.ca

**Russell Greiner**
Department of Computing Science
University of Alberta
Edmonton, AB T6G 2E8
rgreiner@ualberta.ca

## Abstract

The cutting plane method is an augmentative constrained optimization procedure that is often used with continuous-domain optimization techniques such as linear and convex programs. We investigate the viability of a similar idea within message passing – for integral solutions in the context of two combinatorial problems: 1) For Traveling Salesman Problem (TSP), we propose a factor-graph based on Held-Karp formulation, with an exponential number of constraint factors, each of which has an exponential but sparse tabular form. 2) For graph-partitioning (*a.k.a.* community mining) using modularity optimization, we introduce a binary variable model with a large number of constraints that enforce formation of cliques. In both cases we are able to derive simple message updates that lead to competitive solutions on benchmark instances. In particular for TSP we are able to find near-optimal solutions in the time that empirically grows with $N^3$, demonstrating that augmentation is practical and efficient.

## 1 Introduction

Probabilistic Graphical Models (PGMs) provide a principled approach to approximate constraint optimization for NP-hard problems. This involves a message passing procedure (such as max-product Belief Propagation; BP) to find an approximation to *maximum a posteriori* (MAP) solution. Message passing methods are also attractive as they are easily mass parallelize. This has contributed to their application in approximating many NP-hard problems, including constraint satisfaction [1, 2], constrained optimization [3, 4], min-max optimization [5], and integration [6].

The applicability of PGMs to discrete optimization problems is limited by the size and number of factors in the factor-graph. While many recent attempts have been made to reduce the complexity of message passing over high-order factors [7, 8, 9], to our knowledge no published result addresses the issues of dealing with large number of factors. We consider a scenario where a large number of factors represent hard constraints and ask *whether it is possible to find a feasible solution by considering only a small fraction of these constraints.*

The idea is to start from a PGM corresponding to a tractible subsset of constraints, and after obtaining an approximate MAP solution using min-sum BP, augment the PGM with the set of constraints that are violated in the current solution. This general idea has been extensively studied under the

term *cutting plane methods* in different settings. Dantzig *et al.* [10] first investigated this idea in the context of TSP and Gomory *et al.*[11] provided a elegant method to generate violated constraints in the context of finding integral solutions to linear programs (LP). It has since been used to also solve a variety of nonlinear optimization problems. In the context of PGMs, Sontag and Jaakkola use cutting plane method to iteratively tighten the marginal polytope – that enforces the local consistency of marginals – in order to improve the variational approximation [12]. This differs from our approach, where the augmentation changes the factor-graph (*i.e.*, the inference problem) rather than improving the approximation of inference.

Recent studies show that message passing can be much faster than LP in finding approximate MAP assignments for structured optimization problems [13]. This further motivates our inquiry regarding the viability of augmentation for message passing. We present an affirmative answer to this question in application to two combinatorial problems. Section 2 introduces our factor-graph formulations for Traveling Salesman Problem (TSP) and graph-partitioning. Section 3 derives simple message update equations for these factor-graphs and reviews our augmentation scheme. Finally, Section 4 presents experimental results for both applications.

## 2   Background and Representation

Let $x = \{x_1, \ldots, x_D\} \in \mathcal{X} = \mathcal{X}_1 \times \mathcal{X}_2 \ldots \times \mathcal{X}_D$ denote an instance of a tuple of discrete variables. Let $x_\mathcal{I}$ refer to a sub-tuple, where $\mathcal{I} \subseteq \{1, \ldots, D\}$ indexes a subset of these variables. Define the energy function $f(x) \triangleq \sum_{\mathcal{I} \in \mathcal{F}} f_\mathcal{I}(x_\mathcal{I})$ where $\mathcal{F}$ denotes the set of factors. Here the goal of inference is to find an assignment with minimum energy $x^* = \arg_x \min f(x)$. This model can be conveniently represented using a bipartite graph, known as factor-graph [14], where a factor node $f_\mathcal{I}(x_\mathcal{I})$ is connected to a variable node $x_i$ *iff* $i \in \mathcal{I}$.

### 2.1   Traveling Salesman Problem

A Traveling Salesman Problem (TSP) seeks the minimum length tour of $N$ cities that visits each city exactly once. TSP is $\mathcal{NP}$-hard, and for general distances, no constant factor approximation to this problem is possible [15]. The best known exact solver, due to Held *et al.*[16], uses dynamic programming to reduce the cost of enumerating all orderings from $\mathcal{O}(N!)$ to $\mathcal{O}(N^2 2^N)$. The development of many (now) standard optimization techniques, such as simulated annealing, mixed integer linear programming, dynamic programming, and ant colony optimization are closely linked with advances in solving TSP. Since Dantzig *et al.*[10] manually applied the cutting plane method to 49-city problem, a combination of more sophisticated cuts, used with branch-and-bound techniques [17], has produced the state-of-the-art TSP-solver, Concorde [18]. Other notable results on very large instances have been reported by LinKernighan heuristic [19] that continuously improves a solution by exchanging nodes in the tour. In a related work, Wang *et al.*[20] proposed a message passing solution to TSP. However their method does not scale beyond small toy problems (authors experimented with $N = 5$ cities). For a readable historical background of the state-of-the-art in TSP and its various applications, see [21].

#### 2.1.1   TSP Factor-Graph

Let $\mathcal{G} = (\mathcal{V}, \mathcal{E})$ denote a graph, where $\mathcal{V} = \{v_1, \ldots, v_N\}$ is the set of nodes and the set of edges $\mathcal{E}$ contains $e_{i-j}$ *iff* $v_i$ and $v_j$ are connected. Let $x = \{x_{e_1}, \ldots, x_{e_M}\} \in \mathcal{X} = \{0, 1\}^M$ be a set of binary variables, one for each edge in the graph (*i.e.*, $M = |\mathcal{E}|$) where we will set $x_{e_m} = 1$ *iff* $e_m$ is in the tour. For each node $v_i$, let $\partial v_i = \{e_{i-j} \mid e_{i-j} \in \mathcal{E}\}$ denote the edges adjacent to $v_i$. Given a distance function $d : \mathcal{E} \to \Re$, define the **local factors** for each edge $e \in \mathcal{E}$ as $f_e(x_e) = x_e\, d(e)$ – so this is either $d(e)$ or zero. Any valid tour satisfies the following necessary and sufficient constraints – *a.k.a.* Held-Karp constraints [22]:

**1. Degree constraints:** Exactly two edges that are adjacent to each vertex should be in the tour. Define the factor $f_{\partial v_i}(x_{\partial v_i}) : \{0, 1\}^{|\partial v_i|} \to \{0, \infty\}$ to enforce this constraint

$$f_{\partial v_i}(x_{\partial v_i}) \triangleq \mathbb{I}_\infty \left( \sum_{e \in \partial v_i} x_e = 2 \right) \qquad \forall v_i \in \mathcal{V}$$

where $\mathbb{I}_\infty(condition) \triangleq 0$ *iff* the condition is satisfied and $+\infty$ otherwise.

**2. Subtour constraints:** Ensure that there are no short-circuits – *i.e.*, there are no loops that contain strict subsets of nodes. To enforce this, for each $\mathcal{S} \subset \mathcal{V}$, define $\delta(\mathcal{S}) \triangleq \{e_{i-j} \in \mathcal{E} \mid v_i \in \mathcal{S}, v_j \notin \mathcal{S}\}$ to be the set of edges, with one end in $\mathcal{S}$ and the other end in $\mathcal{V} \setminus \mathcal{S}$.

We need to have at least two edges leaving each subset $\mathcal{S}$. The following set of factors enforce these constraints

$$f_{\delta(\mathcal{S})}(x_{\delta(\mathcal{S})}) = \mathbb{I}_\infty \left( \sum_{x_e \in \mathcal{S}} x_e \geq 2 \right) \qquad \forall \mathcal{S} \subset \mathcal{V}, \ \mathcal{S} \neq \emptyset$$

These three types of factors define a factor-graph, whose minimum energy configuration is the smallest tour for TSP.

## 2.2 Graph Partitioning

Graph partitioning –*a.k.a.* community mining– is an active field of research that has recently produced a variety of community detection methods (*e.g.*, see [23] and its references), a notable one of which is Modularity maximization [24]. However, exact optimization of Modularity is $\mathcal{NP}$-hard [25]. Modularity is closely related to fully connected Potts graphical models [26]. However, due to full connectivity of PGM, message passing is not able to find good solutions. Many have proposed various other heuristics for modularity optimization [27, 28, 26, 29, 30]. We introduce a factor-graph representation of this problem that has a large number of factors. We then discuss a stochastic but sparse variation of modularity that enables us to efficiently partition relatively large sparse graphs.

### 2.2.1 Clustering Factor-Graph

Let $\mathcal{G} = (\mathcal{V}, \mathcal{E})$ be a graph, with a weight function $\widetilde{\omega} : \mathcal{V} \times \mathcal{V} \to \Re$, where $\widetilde{\omega}(v_i, v_j) \neq 0$ iff $e_{i:j} \in \mathcal{E}$. Let $Z = \sum_{v_1, v_2 \in \mathcal{V}} \widetilde{\omega}(v_1, v_2)$ and $\omega(v_i, v_j) \triangleq \frac{\widetilde{\omega}}{2Z}$ be the normalized weights. Also let $\omega(\partial v_i) \triangleq \sum_{v_j} \omega(v_i, v_j)$ denote the normalized degree of node $v_i$. Graph clustering using modularity optimization seeks a partitioning of the nodes into unspecified number of clusters $\mathcal{C} = \{\mathcal{C}_1, \dots, \mathcal{C}_K\}$, maximizing

$$q(\mathcal{C}) = \sum_{\mathcal{C}_i \in \mathcal{C}} \sum_{v_i, v_j \in \mathcal{C}_i} \left( \omega(v_i, v_j) - \omega(\partial v_i) \omega(\partial v_j) \right) \tag{1}$$

The first term of modularity is proportional to within-cluster edge-weights. The second term is proportional to the expected number of within cluster edge-weights for a null model with the same weighted node degrees for each node $v_i$.

Here the null model is a fully-connected graph. We generate a random *sparse null model* with $M_{null} < \alpha M$ weighted edges ($\mathcal{E}_{null}$), by randomly sampling two nodes, each drawn independently from $\mathbb{P}(v_i) \propto \sqrt{\omega(\partial v_i)}$, and connecting them with a weight proportional to $\widetilde{\omega}_{null}(v_i, v_j) \propto \sqrt{\omega(\partial v_i)\omega(\partial v_j)}$. If they have been already connected, this weight is added to their current weight. We repeat this process $\alpha M$ times, however since some of the edges are repeated, the total number of edges in the null model may be under $\alpha M$. Finally the normalized edge-weight in the sparse null model is $\omega_{null}(v_i, v_j) \triangleq \frac{\widetilde{\omega}_{null}(v_i, v_j)}{2\sum_{v_i, v_j} \widetilde{\omega}_{null}(v_i, v_j)}$. It is easy to see that this generative process in expectation produces the fully connected null model.[1]

Here we use the following binary-valued factor-graph formulation. Let $x = \{x_{i_1:j_1}, \dots, x_{i_L:j_L}\} = \{0, 1\}^L$ be a set of binary variables, one for each edge $e_{i:j} \in \mathcal{E} \cup \mathcal{E}_{null}$ – *i.e.*, $|\mathcal{E} \cup \mathcal{E}_{null}| = L$. Define the **local factor** for each variable as $f_{i:j}(x_{i:j}) = -x_{i-j}(\omega(v_i, v_j) - \omega_{null}(v_i, v_j))$. The idea is to enforce formation of cliques, while minimizing the sum of local factors. By doing so the

negative sum of local factors evaluates to modularity (eq 1). For each three edges $e_{i:j}, e_{j:k}, e_{i:k} \in \mathcal{E} \cup \mathcal{E}_{null}, i < j < k$ that form a triangle, define a **clique constraint** as

$$f_{\{i:j,j:k,i:k\}}(x_{i:j}, x_{j:k}, x_{i:k}) \triangleq \mathbb{I}_{\infty}(x_{i:j} + x_{j:k} + x_{i:k} \neq 2)$$

These factors ensure the formation of cliques – *i.e.*, if the weights of two edges that are adjacent to the same node are non-zero, the third edge in the triangle should also have non-zero weight. The computational challenge here is the large number of clique constraints. Brandes *et al.*[25] use a similar LP formulation. However, since they include all the constraints from the beginning and the null model is fully connected, their method is only applied to small toy problems.

## 3 Message Passing

Min-sum belief propagation is an inference procedure, in which a set of messages are exchanged between variables and factors. The factor-to-variable ($\nu_{\mathcal{I} \to e}$) and variable-to-factor ($\nu_{e \to \mathcal{I}}$) messages are defined as

$$\nu_{e \to \mathcal{I}}(x_e) \quad \triangleq \quad \sum_{\mathcal{I}' \ni e, \mathcal{I}' \neq \mathcal{I}} \nu_{\mathcal{I}' \to e}(x_e) \tag{2}$$

$$\nu_{\mathcal{I} \to e}(x_e) \quad \triangleq \quad \min \left\{ f_{\mathcal{I}}(x_{\mathcal{I} \setminus e}, x_e) \sum_{e' \in \mathcal{I} \setminus e} \nu_{e' \to \mathcal{I}}(x_{e'}) \right\}_{x_{\mathcal{I} \setminus e}} \tag{3}$$

where $\mathcal{I} \ni e$ indexes all factors that are adjacent to the variable $x_e$ on the factor-graph. Starting from an initial set of messages, this recursive update is performed until convergence.

This procedure is exact on trees, factor-graphs with single cycle as well as some special settings [4]. However it is found to produce good approximations in general loopy graphs. When BP is exact, the set of local beliefs $\mu_e(x_e) \triangleq \sum_{\mathcal{I} \ni e} \nu_{\mathcal{I} \to e}(x_e)$ indicate the minimum value that can be obtained for a particular assignment of $x_e$. When there are no ties, the joint assignment $x^*$, obtained by minimizing individual local beliefs, is optimal.

When BP is not exact or the marginal beliefs are tied, a **decimation** procedure can improve the quality of final assignment. Decimation involves fixing a subset of variables to their most biased values, and repeating the BP update. This process is repeated until all variables are fixed.

Another way to improve performance of BP when applied to loopy graphs is to use **damping**, which often prevents oscillations: $\nu_{\mathcal{I} \to e}(x_e) = \lambda \widetilde{\nu}_{\mathcal{I} \to e}(x_e) + (1 - \lambda) \nu_{\mathcal{I} \to e}(x_e)$. Here $\widetilde{\nu}_{\mathcal{I} \to e}$ is the new message as calculated by eq 3 and $\lambda \in (0, 1]$ is the damping parameter. Damping can also be applied to variable-to-factor messages.

When applying BP equations eqs 2, 3 to the TSP and clustering factor-graphs, as defined above, we face two computational challenges: (a) Degree constraints for TSP can depend on $N$ variables, resulting in $\mathcal{O}(2^N)$ time complexity of calculating factor-to-variable messages. For subtour constraints, this is even more expensive as $f_{\mathcal{S}}(x_{\delta(\mathcal{S})})$ depends on $\mathcal{O}(M)$ (recall $M = |\mathcal{E}|$ which can be $\mathcal{O}(N^2)$) variables. (b) The complete TSP factor-graph has $\mathcal{O}(2^N)$ subtour constraints. Similarly the clustering factor-graph can contain a large number of clique constraints. For the fully connected null model, we need $\mathcal{O}(N^3)$ such factors and even using the sparse null model – assuming a random edge probability *a.k.a.* Erdos-Reny graph – there are $\mathcal{O}(\frac{L^3}{N^6} N^3) = \mathcal{O}(\frac{L^3}{N^3})$ triangles in the graph (recall that $L = |\mathcal{E} \cup \mathcal{E}_{null}|$). In the next section, we derive the compact form of BP messages for both problems. In the case of TSP, we show how to exploit the sparsity of degree and subtour constraints to calculate the factor-to-variable messages in $\mathcal{O}(N)$ and $\mathcal{O}(M)$ respectively.

### 3.1 Closed Form of Messages

For simplicity we work with normalized message $\nu_{\mathcal{I} \to e} \triangleq \nu_{\mathcal{I} \to e}(1) - \nu_{\mathcal{I} \to e}(0)$, which is equivalent to assuming $\nu_{\mathcal{I} \to e}(0) = 0 \ \forall \mathcal{I}, e$. The same notation is used for variable-to-factor message, and marginal belief. We refer to the normalized marginal belief, $\mu_e = \mu_e(1) - \mu(0)_e$ as bias.

Despite their exponentially large tabular form, both degree and subtour constraint factors for TSP are sparse. Similar forms of factors is studied in several previous works [7, 8, 9]. By calculating

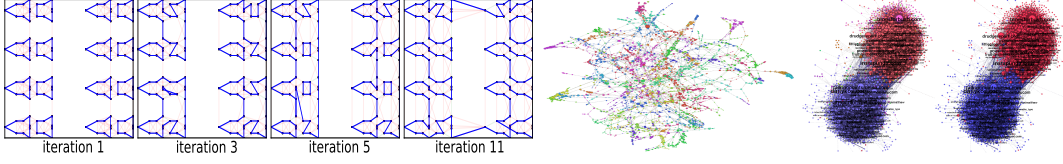

iteration 1    iteration 3    iteration 5    iteration 11

Figure 1: (left) The message passing results after each augmentation step for the complete graph of printing board instance from [31]. The blue lines in each figure show the selected edges at the end of message passing. The pale red lines show the edges with the bias that, although negative ($\mu_e < 0$), were close to zero. (middle) Clustering of power network ($N = 4941$) by message passing. Different clusters have different colors and the nodes are scaled by their degree. (right) Clustering of politician blogs network ($N = 1490$) by message passing and by meta-data – *i.e.*, liberal or conservative.

the closed form of these messages for **TSP factor-graph**, we observe that they have a surprisingly simple form. Rewriting eq 3 for *degree constraint* factors, we get:

$$\nu_{\partial v_i \to e}(1) = \min\{\nu_{e' \to \partial v_i}\}_{e' \in \partial v_i \backslash e} \quad , \quad \nu_{\partial v_i \to e}(0) = \min\{\nu_{e' \to \partial v_i} + \nu_{e'' \to \partial v_i}\}_{e', e'' \in \partial v_i \backslash e} \quad (4)$$

where we have dropped the summation and the factor from eq 3. For $x_e = 1$, in order to have $f_{\partial v_i}(x_{\partial i}) < \infty$, only *one* other $x_{e'} \in x_{\partial v_i}$ should be non-zero. On the other hand, we know that messages are normalized such that $\nu_{e \to \partial v_i}(0) = 0 \;\; \forall v_i, e \in \partial v_i$, which means they can be ignored in the summation. For $x_e = 0$, in order to satisfy the constraint factor, *two* of the adjacent variables should have a non-zero value. Therefore we seek two such incoming messages with minimum values. Let $\min[k]\mathcal{A}$ denote the $k^{th}$ smallest value in the set $\mathcal{A}$ – *i.e.*, $\min \mathcal{A} \equiv \min[1]\mathcal{A}$. We combine the updates above to get a "normalized message", $\nu_{\partial v_i \to e}$, which is simply the negative of the second largest incoming message (excluding $\nu_{e \to \partial v_i}$) to the factor $f_{\partial v_i}$:

$$\nu_{\partial v_i \to e} = \nu_{\partial v_i \to e}(1) - \nu_{\partial v_i \to e}(0) = -\min[2]\{\nu_{e' \to \partial v_i}\}_{e' \in \partial v_i \backslash e} \quad (5)$$

Following a similar procedure, factor-to-variable messages for *subtour constraints* is given by

$$\nu_{\delta(\mathcal{S}) \to e} = -\max\{0, \min[2]\{\nu_{e' \to \delta(\mathcal{S})}\}_{e' \in \delta(\mathcal{S}) \backslash e}\} \quad (6)$$

Here while we are searching for the minimum incoming message, if we encounter two messages with negative or zero values, we can safely assume $\nu_{\delta(\mathcal{S}) \to e} = 0$, and stop the search. This results in significant speedup in practice. Note that both eq 5 and eq 6 only need to calculate the second smallest message in the set $\{\nu_{e' \to \delta(\mathcal{S})}\}_{e' \in \delta(\mathcal{S}) \backslash e}$. In the asynchronous calculation of messages, this minimization should be repeated for each outgoing message. However in a **synchronous update** by finding three smallest incoming messages to each factor, we can calculate all the factor-to-variable messages at the same time.

For the **clustering factor-graph**, the clique factor is satisfied only if either zero, one, or all three of the variables in its domain are non-zero. The factor-to-variable messages are given by

$$\nu_{\{i:j,j:k,i:k\} \to i:j}(0) = \min\{0, \; \nu_{j:k \to \{i:j,j:k,i:k\}}, \; \nu_{i:k \to \{i:j,j:k,i:k\}}\}$$
$$\nu_{\{i:j,j:k,i:k\} \to i:j}(1) = \min\{0, \; \nu_{j:k \to \{i:j,j:k,i:k\}} \; + \; \nu_{i:k \to \{i:j,j:k,i:k\}}\} \quad (7)$$

For $x_{i:j} = 0$, the minimization is over three feasible cases (a) $x_{j:k} = x_{i:k} = 0$, (b) $x_{j:k} = 1, x_{i:k} = 0$ and (c) $x_{j:k} = 0, x_{i:k} = 1$. For $x_{i:j} = 1$, there are two feasible cases (a) $x_{j:k} = x_{i:k} = 0$ and (b) $x_{j:k} = x_{i:k} = 1$. Normalizing these messages we have

$$\nu_{\{i:j,j:k,i:k\} \to i:j} = \min\{0, \; \nu_{j:k \to \{i:j,j:k,i:k\}} \; + \; \nu_{i:k \to \{i:j,j:k,i:k\}}\} - \quad (8)$$
$$\min\{0, \; \nu_{j:k \to \{i:j,j:k,i:k\}}, \; \nu_{i:k \to \{i:j,j:k,i:k\}}\}$$

### 3.2 Finding Violations

Due to large number of factors, message passing for the full factor-graph in our applications is not practical. Our solution is to start with a minimal set of constraints. For TSP, we start with no subtour constraints and for clustering, we start with no clique constraint. We then use message passing to find marginal beliefs $\mu_e$ and select the edges with positive bias $\mu_e > 0$.

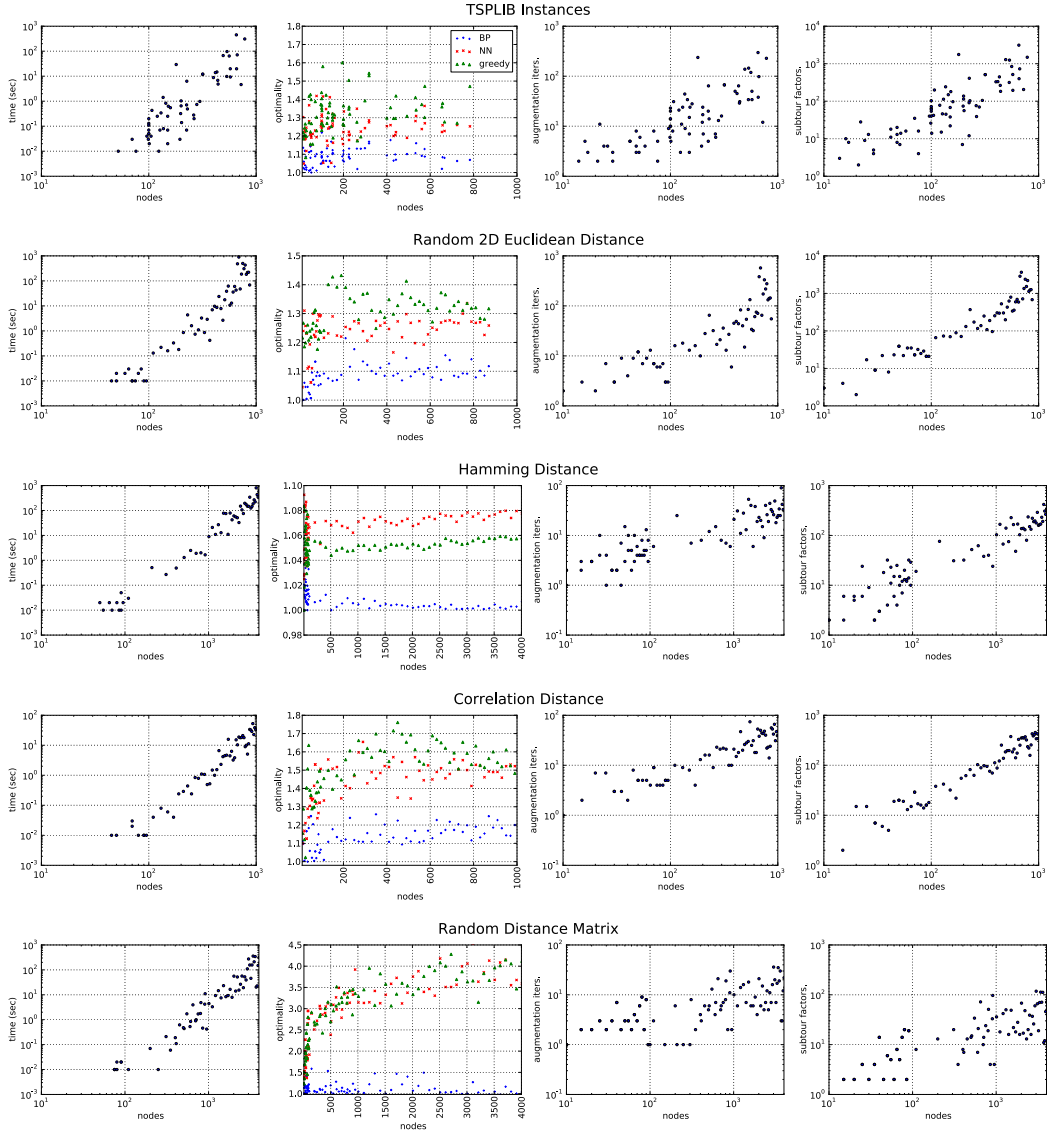

Figure 2: Results of message passing for TSP on different benchmark problems. From left to right, the plots show: (a) running time, (b) optimality ratio (compared to Concorde), (c) iterations of augmentation and (d) number of subtours constraints – all as a function of number of nodes. The optimality is relative to the result reported by Concorde. Note that all plots except optimality are log-log plots where a linear trend shows a monomial relation ($y = ax^m$) between the values on the $x$ and $y$ axis, where the slope shows the power $m$.

We then find the constraints that are violated. For TSP, this is achieved by finding connected components $\mathcal{C} = \{\mathcal{S}_i \subset \mathcal{V}\}$ of the solution in $\mathcal{O}(N)$ time and define new subtour constraints for each $\mathcal{S}_i \in \mathcal{C}$ (see Figure 1(left)).

For graph partitioning, we simply look at pairs of positively fixed edges around each node and if the third edge of the triangle is not positively fixed, we add the corresponding clique factor to the factor-graph; see Appendix A for more details.

## 4 Experiments
### 4.1 TSP
Here we evaluate our method over five benchmark datasets: **(I)** TSPLIB, which contains a variety of real-world benchmark instances, the majority of which are 2D or 3D Euclidean or geographic

Table 1: Comparison of different modularity optimization methods.

| Problem | Weighted? | Nodes | Edges | message passing (full) | | | | message passing (sparse) | | | | Spin-glass | | L-Eigenvector | | FastGreedy | | Louvian | |
|---|---|---|---|---|---|---|---|---|---|---|---|---|---|---|---|---|---|---|---|
| | | | | $L$ | Cost | Modularity | Time | $L$ | Cost | Modularity | Time | Modularity | Time | Modularity | Time | Modularity | Time | Modularity | Time |
| polbooks | y | 105 | 441 | 5461 | 5.68% | 0.511 | .07 | 3624 | 13.55% | 0.506 | .04 | 0.525 | 1.648 | 0.467 | 0.179 | 0.501 | 0.643 | 0.489 | 0.03 |
| football | y | 115 | 615 | 6554 | 27.85% | 0.591 | 0.41 | 5635 | 17.12% | 0.594 | 0.14 | 0.601 | 0.87 | 0.487 | 0.151 | 0.548 | 0.08 | 0.602 | 0.019 |
| wkarate | n | 34 | 78 | 562 | 12.34% | 0.431 | 0 | 431 | 15.14% | 0.401 | 0 | 0.444 | 0.557 | 0.421 | 0.095 | 0.410 | 0.085 | 0.443 | 0.027 |
| netscience | n | 1589 | 2742 | NA | NA | NA | NA | 53027 | **.0004**% | 0.941 | 2.01 | 0.907 | 8.459 | 0.889 | 0.303 | 0.926 | 0.154 | 0.948 | 0.218 |
| dolphins | y | 62 | 159 | 1892 | 14.02% | 0.508 | 0.01 | 1269 | 6.50% | 0.521 | 0.01 | 0.523 | 0.728 | 0.491 | 0.109 | 0.495 | 0.107 | 0.517 | 0.011 |
| lesmis | n | 77 | 254 | 2927 | 5.14% | 0.531 | 0 | 1601 | 1.7% | 0.534 | 0.01 | 0.529 | 1.31 | 0.483 | 0.081 | 0.472 | 0.073 | 0.566 | 0.011 |
| celegansneural | n | 297 | 2359 | 43957 | 16.70% | 0.391 | 10.89 | 21380 | 3.16% | 0.404 | 2.82 | 0.406 | 5.849 | 0.278 | 0.188 | 0.367 | 0.12 | 0.435 | 0.031 |
| polblogs | y | 1490 | 19090 | NA | NA | NA | NA | 156753 | **.14**% | 0.411 | 32.75 | 0.427 | 67.674 | 0.425 | 0.33 | 0.427 | 0.305 | 0.426 | 0.099 |
| karate | y | 34 | 78 | 562 | 14.32% | 0.355 | 0 | 423 | 17.54% | 0.390 | 0 | 0.417 | 0.531 | 0.393 | 0.086 | 0.380 | 0.079 | 0.395 | 0.009 |

distances.[2] **(II)** Euclidean distance between random points in 2D. **(III)** Random (symmetric) distance matrices. **(IV)** Hamming distance between random binary vectors with fixed length (20 bits). This appears in applications such as data compression [32] and radiation hybrid mapping in genomics [33]. **(V)** Correlation distance between random vectors with 5 random features (*e.g.*, using TSP for gene co-clustering [34]). In producing random points and features as well as random distances (in (III)), we used uniform distribution over $[0, 1]$.

For each of these cases, we report the (a) run-time, (b) optimality, (c) number of iterations of augmentation and (d) number of subtour factors at the final iteration. In all of the experiments, we use Concorde [18] with its default settings to obtain the optimal solution.[3] Since there are very large number of TSP solvers, comparison with any particular method is pointless. Instead we evaluate the quality of message passing against the "optimal" solution. The results in Figure 2(2nd column from left) reports the optimality ratio – *i.e.*, ratio of the tour found by message passing, to the optimal tour. To demonstrate the non-triviality of these instance, we also report the optimality ratio for two heuristics that have optimality guarantees for metric instances [35]: (a) *nearest neighbour* heuristic ($\mathcal{O}(N^2)$), which incrementally adds the to any end of the current path the closest city that does not form a loop; (b) *greedy* algorithm ($\mathcal{O}(N^2 \log(N))$), which incrementally adds a lowest cost edge to the current edge-set, while avoiding subtours.

In all experiments, we used the full graph $\mathcal{G} = (\mathcal{V}, \mathcal{E})$, which means each iteration of message passing is $\mathcal{O}(N^2\tau)$, where $\tau$ is the number of subtour factors. All experiments use $T_{\max} = 200$ iterations, $\epsilon_{\max} = \text{median}\{d(e)\}_{e \in \mathcal{E}}$ and damping with $\lambda = .2$. We used decimation, and fixed 10% of the remaining variables (out of $N$) per iteration of decimation.[4] This increases the cost of message passing by an $\mathcal{O}(\log(N))$ multiplicative factor, however it often produces better results.

All the plots in Figure 2, except for the second column, are in log-log format. When using log-log plot, a linear trend shows a monomial relation between $x$ and $y$ axes – *i.e.*, $y = ax^m$. Here $m$ indicates the slope of the line in the plot and the intercept corresponds to $\log(a)$. By studying the slope of the linear trend in the run-time (left column) in Figure 2, we observe that, for almost all instances, message passing seems to grow with $N^3$ (*i.e.*, slope of $\sim 3$). Exceptions are TSPLIB instances, which seem to pose a greater challenge, and random distance matrices which seem to be easier for message passing. A similar trend is suggested by the number of subtour constraints and iterations of augmentation, which has a slope of $\sim 1$, suggesting a linear dependence on $N$. Again the exceptions are TSPLIB instances that grow faster than $N$ and random distance matrices that seem to grow sub-linearly.[5] Finally, the results in the second column suggests that *message passing is able to find near optimal (in average $\sim 1.1$-optimal) solutions for almost all instances and the quality of tours does not degrade with increasing number of nodes.*

### 4.2 Graph Partitioning

For graph partitioning, we experimented with a set of classic benchmarks[6]. Since the optimization criteria is modularity, we compared our method only against best known "modularity optimization" heuristics: (a) FastModularity[27], (b) Louvain [30], (c) Spin-glass [26] and (d) Leading eigenvector [28]. For message passing, we use $\lambda = .1$, $\epsilon_{\max} = median\{|\omega(e) - \omega_{null}(e)|\}_{e \in \mathcal{E} \cup \mathcal{E}_{null}}$ and $T_{\max} = 10$. Here we do not perform any decimation and directly fix the variables based on their bias $\mu_e > 0 \Leftrightarrow x_e = 1$.

Table 1 summarizes our results (see also Figure 1(middle,right)). Here for each method and each data-set, we report the *time* (in seconds) and the *Modularity* of the communities found by each method. The table include the results of message passing for both full and sparse null models, where we used a constant $\alpha = 20$ to generate our stochastic sparse null model. For message passing, we also included $L = |\mathcal{E} + \mathcal{E}_{null}|$ and the saving in the *cost* using augmentation. This column shows the percentage of the number of all the constraints considered by the augmentation. For example, the cost of `.14%` for the polblogs data-set shows that augmentation and sparse null model meant using `.0014` times fewer clique-factors, compared to the full factor-graph.

Overall, the results suggest that our method is comparable to state-of-the-art in terms both time and quality of clustering. But more importantly it shows that augmentative message passing is able to find feasible solutions using a small portion of the constraints.

## 5 Conclusion

We investigate the possibility of using cutting-plane-like, augmentation procedures with message passing. We used this procedure to solve two combinatorial problems; TSP and modularity optimization. In particular, our polynomial-time message passing solution to TSP often finds near-optimal solutions to a variety of benchmark instances.

Despite losing the guarantees that make cutting plane method very powerful, our approach has several advantages: First, message passing is more efficient than LP for structured optimization [13] and it is highly parallelizable. Moreover by directly obtaining integral solutions, it is much easier to find violated constraints. (Note the cutting plane method for combinatorial problems operates on *fractional* solutions, whose rounding may eliminate its guarantees.) For example, for TSPs, our method simply adds violated constraints by finding connected components. However, due to non-integral assignments, cutting plane methods require sophisticated tricks to find violations [21]. Although powerful branch-and-cut methods, such as Concorde, are able to exactly solve instances with few thousands of variables, their general run-time on benchmark instances remains exponential [18, p495], while our approximation appears to be $\mathcal{O}(N^3)$. Overall our studies indicate that augmentative message passing is an efficient procedure for constraint optimization with large number of constraints.

## Footnotes

[1]The choice of using square root of weighted degrees for both sampling and weighting is to reduce the variance. One may also use pure importance sampling (*i.e.*, use the product of weighted degrees for sampling and set the edge-weights in the null model uniformly), or uniform sampling of edges, where the edge-weights of the null model are set to the product of weighted degrees.

[2]Geographic distance is the distance on the surface of the earth as a large sphere.

[3]For many larger instances, Concorde (with default setting and using CPLEX as LP solver) was not able to find the optimal solution. Nevertheless we used the upper-bound on the optimal produced by Concord in evaluating our method.

[4]Note that here we are only fixing the top $N$ variables with *positive* bias. The remaining $M - N$ variables are automatically clamped to zero.

[5]Since we measured the time in milliseconds, the first column does not show the instances that had a running time of less than a millisecond.

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
