[Supplementary Material]

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

**input** : Graph $\mathcal{G} = (\mathcal{V}, \mathcal{E})$, distance function $d : \mathcal{E} \to \Re$, maximum iterations $T_{max}$, damping $\lambda$,
       threshold $\epsilon_{\max}$.
**output**: A subset $\mathcal{T} \subset \mathcal{E}$ of the edges in the tour.
construct the initial factor-graph
initialize the messages $\nu_{i \to e} \leftarrow 0 \;\; \forall i, e \in \partial i$
initialize $\mu_e \leftarrow d(e) \;\; \forall e \in \mathcal{E}$
**while** *True* **do** // the augmentation loop

    $\epsilon \leftarrow 0, T \leftarrow 0$
    **while** $\epsilon < \epsilon_{\max}$ *and* $T < T_{\max}$ **do** // BP loop

        $\epsilon \leftarrow 0$
        **for** *each* $f(x_{\mathcal{I}})$ **do** // inc. $f_{\delta(\mathcal{S})}$, $f_{\partial v_i}$

            find three lowest values in $\{\nu_{e \to \mathcal{I}}\}_{e \in \mathcal{I}}$
            **for** *each* $e \in \mathcal{I}$ **do**
                calculate $\widetilde{\nu}_{\mathcal{I} \to e}$ using eqs (5,6)
                $\epsilon_{\mathcal{I} \to e} \leftarrow \widetilde{\nu}_{\mathcal{I} \to e} - \nu_{\mathcal{I} \to e}$
                $\nu_{\mathcal{I} \to e} \leftarrow \nu_{\mathcal{I} \to e} + \lambda \epsilon_{\mathcal{I} \to e}$
                $\mu_e \leftarrow \mu_e + \epsilon_{\mathcal{I} \to e}$
                $\epsilon \leftarrow \max\{\epsilon, |\epsilon_{\mathcal{I} \to e}|\}$
            **end**

        **end**
        $T \leftarrow T + 1$
    **end**
    $\mathcal{T} \leftarrow \{e \in \mathcal{E} \mid \mu_e > 0\}$                 // respecting degree constraints.
    $\mathcal{C} \leftarrow$ ConnectedComponents$((\mathcal{V}, \mathcal{T}))$
    **if** $|\mathcal{C}| = 1$ **then** return $\mathcal{T}$
    **else** augment the factor-graph with $f_{\mathcal{S}_i}(x_{\delta(\mathcal{S}_i)}) \;\forall \mathcal{S}_i \in \mathcal{C}$
    initialize $\nu_{\mathcal{S}_i \to e} \leftarrow 0 \;\forall \mathcal{S}_i \in \mathcal{C}, e \in \mathcal{S}_i$

**end**

**Algorithm 1:** Message Passing for TSP

**input** : Graph $\mathcal{G} = (\mathcal{V}, \mathcal{E})$, weight function $\widetilde{\omega} : \mathcal{E} \to \Re$, maximum iterations $T_{max}$, damping $\lambda$,
           threshold $\epsilon_{\max}$.
**output**: A clustering $\mathcal{C} = \{\mathcal{C}_1, \ldots, \mathcal{C}_K\}$ of nodes.
construct the null model
$\mu_e \leftarrow 0 \; \forall e \in \mathcal{E} \cup \mathcal{E}_{null}$
**while** *True* **do** // the augmentation loop

> $\epsilon \leftarrow 0, T \leftarrow 0$
> **while** $\epsilon < \epsilon_{\max}$ *and* $T < T_{\max}$ **do** // BP loop
>
> > $\epsilon \leftarrow 0$
> > **for** $e_{i-j} \in \mathcal{E} \cup \mathcal{E}_{null}$ **do**
> >
> > > $\mu_{e_{i-j}}^{old} \leftarrow \mu_{e_{i-j}}$
> > > $\mu_{e_{i-j}} \leftarrow (\omega(v_i, v_j) - \omega_{null}(v_i, v_j))$
> > > **for** $\mathcal{I} \ni e_{i-j}$ **do** // update beliefs
> > >
> > > > calculate $\nu_{\mathcal{I} \to e_{i-j}}$ using eq 8
> > > > $\mu_{e_{i-j}} \leftarrow \mu_{e_{i-j}} + \nu_{\mathcal{I} \to e_{i-j}}$
> > >
> > > **end**
> > > $\epsilon \leftarrow \max\{\epsilon, |\mu_{e_{i-j}} - \mu_{e_{i-j}}^{old}|\}$
> > > **for** $\mathcal{I} \ni e_{i-j}$ **do** // update msgs.
> > >
> > > > $\widetilde{\nu}_{e_{i-j} \to \mathcal{I}} \leftarrow \mu_{e_{i-j}} - \nu_{\mathcal{I} \to e_{i-j}}$
> > > > $\nu_{e_{i-j} \to \mathcal{I}} \leftarrow \lambda \widetilde{\nu}_{e_{i-j} \to \mathcal{I}} + (1 - \lambda) \nu_{e_{i-j} \to \mathcal{I}}$
> > >
> > > **end**
> >
> > **end**
> > $T \leftarrow T + 1$
>
> **end**
> **for** $v_i \in \mathcal{V}$ **do**
>
> > **for** $e_{i-j}, e_{i-k} \in \mathcal{E} \cup \mathcal{E}_{null}$ **do**
> >
> > > **if** $\mu_{e_{i-j}} > 0$ *and* $\mu_{e_{i-k}} > 0$ *and* $\mu_{e_{i-k}} \leq 0$ **then** add the corresponding clique factor to the factor-graph
> >
> > **end**
>
> **end**
> **if** *no factor was added* **then break** out of the loop
> **else** $\nu_{e \to \mathcal{I}} \leftarrow 0 \; \forall \mathcal{I}, e \in \mathcal{I}$

**end**
$\mathcal{C} \leftarrow \texttt{ConnectedComponents}((\mathcal{V}, \{e \in \mathcal{E} \cup \mathcal{E}_{null} \mid \mu_e > 0\}))$

**Algorithm 2:** Message Passing for Modularity Maximization.

# A  Factor-Graphs and PseudoCodes

Algorithms 1 and 2 present the pseudocode for both TSP and graph-partitioning by message passing. Note that the scheduling of message updates in these two algorithms is very different. This difference in scheduling is mainly due to the presense of high-order factors in TSP factor-graph and intends to minimize the time complexity. Also, while TSP message passing is re-using the messages from the previous augmentation iteration, for clustering, we initialize the messages to zero. This is because the number of factors in each augmentation step for clustering is relatively large and in practice initializing the messages to zero is more efficient. For both problems, we have included the message from *local* factors in the marginals and therefore they are ignored during the message update. In practice we do not need to store *any* of the messages for TSP. Instead we can only keep the three smallest incoming messages to each factor and calculate factor-to-variable messages using these values. The variable-to-factor messages can also be recomputed as required using the marginals and factor-to-variable messages: $\nu_{e \to \mathcal{I}}(x_e) = \mu_e(x_e) - \nu_{\mathcal{I} \to e}(x_e)$.