[Reviews · NeurIPS 2014]

Submitted by Assigned_Reviewer_41

Augmentative Message Passing for Traveling Salesman Problem and Graph Partitioning

The paper develops messaging passing techniques to approximate two NP-hard problems : TSP and graph partitioning. Using specifics of each problem a graphical model is constructed such that the MAP state gives the solution for the original problem (e.g. using Held-Karp necessary and sufficient conditions for valid tours for the TSP). The resulting model has a very large number of clusters making message passing on the full graph (full set of constraints) infeasible for large problems (even when loops are ignored in the problem and no junction tree is constructed). The paper proposes methods to approximate this by starting with a smaller set of constraints and iteratively adding some.

Quality
The methods proposed do not come with any guarantee. The development of the algorithm according to the declared principles appears to be correct.

Clarity
The overview of the algorithm in the appendix is a nice addition to the main text.

Originality
This way of trying to approximate TSP and graph partitioning is to the best of my knowledge new. Methods such as survey propagation have been proposed earlier to approximate SAT problems but are significantly different.

Significance
The significance of this method is that it provides an idea of what these type of methods could mean for notoriously difficult optimization problems such as TSP and graph partitioning. Graph partioning gives performance roughly on par with state of the art. The TSP part is compared only to exact. The claim is "Since there are very large number of TSP solvers, comparison with any particular method is pointless." This is not the strongest part of the paper: it would definitely make sense to get a feel if these methods come close to a point on the speed-accuracy pareto curve. But it is not claimed the methods beat state-of-the-art and are stated to be a proof-of-principle.
Summary: A new way of leveraging message passing techniques to approximate two hard optimization problems. The methods do not come with a guarantee but are interesting and get close to or about the same as state-of-the-art.

Submitted by Assigned_Reviewer_42

The paper proposes a procedure for message-passing algorithm that at first considers subset of constraints, and after augments the constraint set with the violated set. The procedure with the resulting message-passing updates was tested on two difficult problems.

I find the paper overall well written and the idea behind the approach very interesting. The only drawback is the lack of analysis, which is probably a difficult task.
Summary: I find the approach creative and interesting with the many possible applications.

Submitted by Assigned_Reviewer_44

The authors studied how to solve NP-hard TSP and graph partition problem as an inference on a graphical model. They first showed how to design a graphical model for each problem. The corresponding graphs are cyclic and the inference problems are NP hard. Because of the success of the belief propagation (BP) method, which provides a good approximate inference with cheap computational cost in error correction codes and related problems, the authors applied min-sum BP for TSP and graph partition problem.

Most part of the paper is clearly written. The approach is interesting, but this is an empirical study. My recommendation is positive but not strong.

The paper has two main claims. One is how to design the graphical models and apply BP algorithm. Naïve BP may not work well, and the authors combined some tricks. The other point is to show the numerical results of min-sum. The authors claim the results are comparable to state-of-art methods but the computations cost is small.

First point: The graphical model representation of TSP is straightforward. That for the graph partition is realized via a stochastic approach, which is not so difficult. The important part is how to apply BP successfully. The graphs have a lot of cycles and naïve min-sum BP may not work well. For each problem, they combined some approaches: the messages are normalized, and the subtour/clique constraints are treated carefully. They also combined and decimation for TSP and damping for both. These are heuristic and we would like to know if they really avoided local maxima and how often the algorithm converges to the optimal points.

Second point: For graphs with many cycles, min-sum BP generally does not provide the optimal solution. We confirm this fact from numerical results of the paper. Although the authors claim the cheap computational cost and the good quality of the solutions, it is difficult to convince non-specialists to use this approach.

Overall comment: The authors have shown numerical results for known data. What people want to know is what will happen for new unknown data. Some theoretical studies of the characteristics should be shown. For what kind of data, this method is better than other methods, and how the computational cost grows according to the size of the problem (the discussion in section 4 is not sufficient).

Finally, I have one question. The graph partitioning method uses a randomized method. How the results are influenced with it?
Summary: Most part of the paper is clearly written. The approach is interesting, but this is an empirical study. My recommendation is positive but not strong.
Author Feedback
Author rebuttal: We thank all 3 reviewers for their feedback. Although the reviews are all favourable, two of the reviews are not very confident about the “potential impact” of our work. Here, we try to clarify some points to improve this view.

To our knowledge, except for two problems of matching [1,2] and minimum spanning tree [3] (both of which are in class P), none of the message passing solutions to combinatorial problems come with any forms of guarantee. Despite the fact that evaluation of message passing methods has otherwise “always” been experimental, they have received a lot of attention and had a high impact (e.g. in clustering [4], satisfiability [5], coloring[6] and steiner tree problem [7]) and tricks such as damping and normalization of messages to simplify updates is often used (e.g. [4,5]). Therefore our results both in terms of evaluation method, guarantees and optimality are similar to previous high-impact works and we look forward to positive feedback from the community.

------------------------------

wrt reviewer (44) questions:

Reviewer: The graph partitioning method uses a randomized method. How the results are influenced with it?

Answer: Table 1. shows that the modularity, using the stochastic sparse model is very slightly worse than the fully connected model (which is still using augmentation), but the sparse version is applicable to larger instances.

Reviewer: The authors have shown numerical results for known data. What people want to know is what will happen for new unknown data.

Authors: It is not clear what “unknown” data is (there is no training or learning involved).

[1] M. Bayati, D. Shah, and M. Sharma. Maximum weight matching via max-product
belief propagation. In ISIT, pages 1763–1767. IEEE, 2005.

[2] B. Huang and T. Jebara. Loopy belief propagation for bipartite maximum weight
b-matching. In AI and Statistics, 2007

[3] M. Bayati, A. Braunstein, and R. Zecchina. A rigorous analysis of the cavity equations
for the minimum spanning tree. Journal of Mathematical Physics, 49(12):125206,
2008

[4] B. Frey and D. Dueck. Clustering by passing messages between data points. Science,
2007

[5] M. Mezard, G. Parisi, and R. Zecchina. Analytic and algorithmic solution of random
satisfiability problems. Science, 297(5582):812–815, Aug. 2002.

[6] A. Braunstein, R. Mulet, A. Pagnani, M. Weigt, and R. Zecchina. Polynomial iterative
algorithms for coloring and analyzing random graphs. Physical Review E, 68(3):
036702, 2003

[7] Bailly-Bechet, Marc, et al. Finding undetected protein associations in cell signaling by belief propagation. Proceedings of the National Academy of Sciences 108.2 (2011): 882-887.